# Preclinical Studies in Small Animals for Advanced Drug Delivery Using Hyperthermia and Intravital Microscopy

**DOI:** 10.3390/cancers13205146

**Published:** 2021-10-14

**Authors:** Marjolein I. Priester, Sergio Curto, Ann L. B. Seynhaeve, Anderson Cruz Perdomo, Mohamadreza Amin, Pierre Agnass, Milad Salimibani, Pegah Faridi, Punit Prakash, Gerard C. van Rhoon, Timo L. M. ten Hagen

**Affiliations:** 1Laboratory Experimental Oncology (LEO), Department of Pathology, Erasmus University Medical Center, 3015 GD Rotterdam, The Netherlands; m.priester@erasmusmc.nl (M.I.P.); a.seynhaeve@erasmusmc.nl (A.L.B.S.); m.amin@erasmusmc.nl (M.A.); 2Department of Radiotherapy, Erasmus MC Cancer Institute, University Medical Center Rotterdam, 3015 GD Rotterdam, The Netherlands; s.curto@erasmusmc.nl (S.C.); a.cruzperdomo@erasmusmc.nl (A.C.P.); p.agnass@amsterdamumc.nl (P.A.); miladsalimibani@gmail.com (M.S.); g.c.vanrhoon@erasmusmc.nl (G.C.v.R.); 3Nanomedicine Innovation Center Erasmus (NICE), Erasmus University Medical Center, 3015 GD Rotterdam, The Netherlands; 4Department of Mechanical Engineering, Iran University of Science and Technology, Tehran 16887, Iran; 5Department of Electrical and Computer Engineering, Kansas State University, Manhattan, KS 66506, USA; pegah@ksu.edu (P.F.); prakashp@ksu.edu (P.P.)

**Keywords:** hyperthermia, preclinical hyperthermia, small animals, multimodal therapy, optical monitoring, dorsal skinfold chamber, radiofrequency hyperthermia, microwave hyperthermia

## Abstract

**Simple Summary:**

Mild hyperthermia is a technique that induces a local tumor temperature increase up to 43.0 °C. In this paper, we introduce three devices that could be used to apply hyperthermia in small animals in combination with intravital microscopy for the real-time visualization of in vivo intratumoral events such as drug delivery.

**Abstract:**

This paper presents three devices suitable for the preclinical application of hyperthermia via the simultaneous high-resolution imaging of intratumoral events. (Pre)clinical studies have confirmed that the tumor micro-environment is sensitive to the application of local mild hyperthermia. Therefore, heating is a promising adjuvant to aid the efficacy of radiotherapy or chemotherapy. More so, the application of mild hyperthermia is a useful stimulus for triggered drug release from heat-sensitive nanocarriers. The response of thermosensitive nanoparticles to hyperthermia and ensuing intratumoral kinetics are considerably complex in both space and time. To obtain better insight into intratumoral processes, longitudinal imaging (preferable in high spatial and temporal resolution) is highly informative. Our devices are based on (i) an external electric heating adaptor for the dorsal skinfold model, (ii) targeted radiofrequency application, and (iii) a microwave antenna for heating of internal tumors. These models, while of some technical complexity, significantly add to the understanding of effects of mild hyperthermia warranting implementation in research on hyperthermia.

## 1. Introduction

Hyperthermia (HT) is an adjuvant treatment modality in which local, partial, or whole-body temperature is elevated above normal physiological value for a predefined period of time. The application of heat is able to induce both a cellular and physiological response, and it could therefore lead to multifactorial therapeutic benefits [1]. However, this is highly dependent on several factors such as the achieved temperature, exposure time, tumor location, thermotolerance of the targeted cells, and tumor vascular bed [2,3]. 

For instance, cellular-level mild hyperthermia (40.0–43.0 °C) can induce direct cytotoxicity via the sensitization of the tumor cell membrane and cytoskeleton [4,5]. Exceeding the physiological temperature threshold could eventually result in programmed cell death or irreversible protein denaturation in tumor cells [1,6,7].

On a physiological level, the application of mild HT can also be beneficial when combined with either radiotherapy or chemotherapy. Selective heating could lead to localized vasodilatation, increased perfusion, and the temporary alteration of vascular integrity [8]. This, in turn, may lead to the improved sensitivity and uptake of anti-cancer treatments, as the vascular alterations are thought to contribute to the normalization of the tumor oxygen, nutrient, and pH levels [9]. However, these changes could simultaneously lead to the development of undesired effects such as erythrocyte extravasation (hemorrhage) [10,11]. Further exploitation of the heated tumor micro-environment can be attained with the usage of chemotherapeutics encapsulated in smart drug delivery systems such as thermosensitive liposomes (TSLs).

Thermosensitive liposomes are lipid-based spherical vesicles that can be triggered to release their cargo under the influence of a specific temperature range [12,13]. These types of nanoparticles are not only advantageous due to their improved pharmacokinetics. TSLs also exhibit enhanced retention in comparison to administration of free drugs [14]. Additionally, in combination with mild hyperthermia, TSLs provide an enhancement in tumor targeting [15]. First, passive targeting relies on the presence of vascular fenestrations, which increase in size upon exposure to local HT [15]. Second, HT acts as a stimulus for locally triggered release into tumor vasculature and interstitium [1,7].

Despite its promising features, application of local hyperthermia in preclinical models in a reliable and reproducible manner is challenging. Classic established heating methods are practical and adequate to determine whether mild HT is able to induce a therapeutic effect [16]. However, for a full understanding of the features of hyperthermia-triggered delivery with TSLs, more in-depth knowledge of the dynamic biological processes occurring before, during, and after hyperthermia treatment is required. The incorporation of intravital imaging has enabled the enhanced spatial and temporal resolution of biological processes occurring in the target area [17,18]. In this paper, the development and application of three advanced preclinical small animal heating systems, which can be operated simultaneously with intravital microscopy (IVM), are described (Figure 1).

The heating devices described here are located externally and allow for the real-time in vivo observation of the targeted tumor. First is the dorsal skinfold chamber (DSC) model, which is suitable for the treatment of superficial tumors. Over the last few decades, several different types of dorsal skinfold chambers have been designed. Initial studies implemented the use of kodaloid, lucite, aluminum, and titanium-based skinfold window chambers [19,20,21,22,23,24]. The DSC frames used in this study were made of PEEK. Second is the system designed for the treatment of internal tumors using radiofrequency (RF)-based deep hyperthermia. Last is the application of local hyperthermia using a microwave (MW) hyperthermia applicator, which is suitable for the treatment of both superficial and internal tumors.

## 2. Materials and Methods

### 2.1. Cell Lines and Culture

B16BL6 and BFS-1 were routinely cultured in Dulbecco’s Modified Eagles’ Medium supplemented with fetal calf serum (FCS; 10%) and grown to 80% confluence prior to use. Single cell suspensions for tumor inoculation were prepared by enzymatic detachment using trypsin-EDTA solution. Viability was assessed by trypan blue exclusion, always exceeding 90%. All reagents were purchased from Sigma-Aldrich (Zwijndrecht, The Netherlands).

### 2.2. Animals

The animal experiments presented in this publication were approved by the committee on Animal Research of the Erasmus MC (Rotterdam, The Netherlands) and conducted with permission granted by the Nederlandse Dierexperimentencommissie. All experiments were performed according to the European directive 2010/63/EU on the protection of animals used for scientific purposes.

C57BL/6 mice were purchased from Envigo (Horst, The Netherlands). C57BL/6 mice with the constitutive vascular endothelial cell expression of an eNOS-Tag-GFP (green fluorescent protein) fusion protein were bred in-house. The animals were housed 3–5 per cage under standard conditions: a temperature of 20.0–22.0 °C, a relative humidity of 50–60%, and a 12 h light/dark cycle. Both acidified vitamin C-fortified water and rodent food were available ad libitum. Mice were at least 10 weeks old and weighed minimally 20 g at the start of the study.

### 2.3. Tumor Model

The animal tumor models were based on the surgical introduction of a small tumor fragment. Surgical procedures were performed under the inhalation of anesthesia, and eye ointment was applied. During this period, the maintenance of the animal’s core body temperature was ensured with the use of a heating plate set at 37.0 °C.

#### 2.3.1. Dorsal Tumor Fragment

The surgical procedure to establish the dorsal skinfold chamber model was performed as described previously by Seynhaeve et al. [25] (Figure 2). During the developmental stage of the tumor, the animal was individually housed (32.0 °C and >70% RH). The environmental temperature and relative humidity were set higher than the values mentioned in Section 2.2 in order to keep the protruding skin flap warm and humid. Experiments started approximately 14 days after implantation. The DSC model allows for longitudinal studies, and at the end of the study, animals were sacrificed via cervical dislocation under anesthesia.

#### 2.3.2. Hepatic Tumor Fragment

The surgical procedure to establish the intrahepatic tumor was performed as previously described [26]. Immediately after surgery, the animal was placed under a heating lamp until fully recovered from anesthesia. During the developmental stage of the tumor, the animal was individually housed under standard conditions as described in Section 2.2. Experiments started approximately 10 days after surgical implantation. Due to the invasive nature of the procedure, the animals were immediately euthanized without recovery after each study by cervical dislocation.

### 2.4. In Vivo Temperature Measurement

The thermographic monitoring of superficial layers during HT exposure can be performed with the use of an infrared camera (T1020, FLIR Systems, Inc., Boston, MA, USA). Thermometry at deeper tissue levels was performed via the insertion of thermocouples. For the dorsal skinfold window model, tissue temperatures were measured with point-welded thin manganese and constantan thermocouples [10]. For the cylindrical RF device, gallium arsenide-based fiber optic sensors (Takaoka Toko Co., Ltd., Tokyo, Japan) were placed in three locations. For the directional microwave antenna model, tissue temperatures were measured in two locations with needle-tip thermocouples (Thermo-Electra, Pijnacker, The Netherlands).

### 2.5. Hyperthermia Set-Up

Animals were exposed to mild hyperthermia using the three described heating devices. The devices were developed in collaboration with the Department of Experimental Medical Instrumentation at the Erasmus Medical Center Rotterdam unless stated otherwise. All hyperthermic and imaging procedures were performed under inhalation anesthesia, and eye ointment was applied.

#### 2.5.1. Dorsal Skinfold Chamber (DSC)

Homogeneous local heat deposition was provided by attaching a circular resistive electric heating element to the window (Figure 2B,C). A schematic diagram for the DSC hyperthermia set-up is shown in Appendix A. Microscope images were acquired with use of a confocal microscope (LSM 510 META, Zeiss, Sliedrecht, The Netherlands).

The intravital microscope set-up of the DSC model was previously described by Seynhaeve et al. [25] (Figure 2D). In brief, the sedated animal was placed in a lateral position on a heated platform to keep the body at a constant temperature (37.0 °C). The DSC was subsequently mounted to the chamber-to-stage holder. The heating element was located underneath the microscope viewing port of the holder, aligning with the dorsal skinfold window.

#### 2.5.2. Cylindrical RF (CRF) Device

The device for deep thermal delivery possessed a cylindrical shape with a hollow bore (Figure 3A). The inclusion of a view port at the top of the device allowed for intravital imaging (SP5, Leica Microsystems, Wetzlar, Germany) (Figure 3B). An oscillating electromagnetic field was generated by semi-circle copper antennas (*n* = 4), operating at a frequency of 433.92 MHz (ISM band), which surrounded the cylinder base on either side. A water bolus, enclosing the body, prevented electromagnetic (EM) distortions by coupling the waves and simultaneously ensured the maintenance of the animal’s physiological temperature [27]. The fluid dynamics within the bolus not only affected the EM waves and the skin surface temperature but also improved the specific absorption rate (SAR). The SAR is defined as a measure of the rate at which energy is absorbed by the body when exposed to a radiofrequency electromagnetic field. The maintenance of a high and uniform flow is crucial because the formation of low flow regions across the bolus prolongs the transition time. This results in the increased absorption of energy from the device by the skin and subsequent hot spot formation [27].

A schematic diagram for the cylindrical RF hyperthermia set-up is shown in Appendix A. In brief, temperature sensors were connected to the power generator. To ensure continuous water circulation throughout the bolus, the RF device was connected to a water pump attached to a temperature-controlled water bath.

To validate the deep hyperthermia device, we designed mouse-shaped torso phantoms (⌀ = 2.6 cm) resembling the dielectric properties of muscle tissue (ε_r_ = 58.0; σ = 0.8 S/m; and f = 433.92 MHz). The tissue-equivalent gel was composed of DI water, salt, poly-ethylene, TXT-151, agar, and formaldehyde [28].

#### 2.5.3. Directional Microwave Antenna (DMA)

The hyperthermia set-up for this device consisted of multiple instruments: a microscope stage, a directional microwave hyperthermia applicator, and a workstation. The microscope stage enabled the maintenance of body temperature, the fixation of anesthesia tubing, and the visualization of the liver tissue using an inverted microscope (LSM 510 META, Zeiss, Sliedrecht, The Netherlands) (Figure 4C).

The directional microwave hyperthermia applicator was previously described by Curto et al. and Faridi et al. [29,30]. In brief, a monopole was implemented with a coaxial cable, as illustrated in Figure 4A. The workstation consisted of multiple machines (Appendix A). The antenna was temperature-controlled with a water bath (Figure 4B). Similar to the cylindrical RF device, the presence of water provided a high-dielectric constant medium and in addition generated active cooling throughout the entire device [29]. RF power was provided by connecting the antenna to an RF amplifier (2450 MHz, PinkRF, Nijmegen, The Netherlands) (Figure 4B). The microwave applicator was placed in the designed holder, covered with ultrasound gel, and positioned to radiate towards the tumor (Figure 4C).

In order to deliver specific and stable heating in the target area, software was developed in-house with the Department of Radiotherapy. The insertion of a needle-tip thermocouple into the tumor tissue provided real-time temperature measurements in the process of achieving the set target temperature of 42.0 °C.

## 3. Results

### 3.1. Dorsal Skinfold Chamber (DSC)

The original calculations of temperature distribution within the dorsal skinfold chamber model were simplified versions. These considered not only a one-dimensional distribution, but also a constant temperature on both sides of the window, and they neglected the occurrence of heat loss in tissue due to perfusion (data not shown). Simulations using SEMCAD X software (SPEAG, Zurich, Switzerland) were performed in order to obtain multidimensional realistic modelling. The DSC model consisted of a standard glass cover slip (blue), the tissue layer (red), and a combination of filler glass and a glass cover slip on the backside (blue) (Figure 5A). The presence of glass ensured the maintenance of the system’s integrity and protected the tissue. By discretizing the structures of the heating device into voxels, temperature profiles could be modeled. Simulating the DSC model using large (0.299 mcel; blue) and small (7.040 mcel; red) voxels yielded similar results (Figure 5B). The center of the ring (point 2) showed a minor drop in temperature in comparison to the edges of the heating element (point 1 and 3). This could be attributed to heat distribution solely based on convection due to the presence of a hollow bore for transillumination (Figure 5C). In the figure, the two peaks (point 1 and 3) represent the edges of the heating element, which encompasses the heat-radiating coil (Figure 5B). As the edges of the element were in direct contact with the lower glass plate, heat distribution occurred through both conduction and convection, resulting in a simulated temperature that approximated the set temperature (40.0 °C).

The temperature profile within the window was dependent on several factors: the thermal conductivity (k), surface area (A), temperature (T), and thickness of the layers (L). In order to reach the tumor tissue, the heat flow encountered two different conductivities as it had to travel through both the glass and tissue layer of the DSC (Figure 6A). Our aim was to reach a tumor tissue temperature of 45.0 °C. SEMCAD X simulations indicated that the set temperature had to be ±47.5 °C in order to approximate the desired tissue temperature at ±44.5 °C (Figure 6B). A thermal steady state within the tumor was reached after approximately 6 min (data not shown). SEMCAD X simulations were also performed with a set-up including two heating rings. The placement of a second ring against the glass layer adjacent to the tumor did not automatically lead to a more uniform heating pattern. Another consideration was that the placement of an aluminum element at the tumor-side of the window could impact the performance of the microscope objective. However, the simulations did indicate that the dual ring set-up required less power to reach the desired tissue temperature. The final design of the external heating element consisted of a single heating ring placed at the bottom of the dorsal skinfold window.

In vivo temperature profiles within the dorsal skinfold chamber were determined by controlling the power generator to obtain tissue temperatures within a range of 39.0–43.0 °C. The HT system allowed for fully controlled heating as it was possible to generate 1.0 °C increments in the tumor tissue temperature (Figure 7A). Upon reaching the set temperature, both tumor (red) and surrounding tissue (pink) temperatures reached a plateau as the undershoot or overshoot did not exceed 0.1 °C for any temperature up to 42.0 °C. A rectal temperature probe (purple) indicated stable core body temperatures throughout heat application. Figure 7B shows that the device was able to induce controlled hyperthermia treatment (T_target_ = 42.0 °C). At the onset of heating, the intratumoral temperature rose relatively quickly to 41.0 °C. Subsequently, the feedback loop ensured a gradual temperature build-up to the target temperature. The discontinuation of heat application immediately induced cool down, and the tumor temperature returned to around 31.0 °C.

The application of mild hyperthermia to a tumor in the dorsal skinfold chamber model was able to trigger both extravasation and drug release in the tumor vasculature. Mice bearing a B16BL6 murine melanoma or a BFS-1 murine fibrosarcoma were injected with rhodamine-labelled long circulating liposomes. Under physiological tumor temperatures within the window (±31.0 °C), no liposome extravasation was observed (Figure 8A). Subsequent heating of the tissue within the window (41.0 °C; 30 min) resulted in extensive liposome extravasation (Figure 8A). This indicates that HT application caused an increase in vascular permeability, subsequently resulting in liposome extravasation. Mice bearing a B16BL6 murine melanoma were injected with carboxyfluorescein-loaded thermosensitive liposomes (Figure 8B). Starting at an ambient temperature (±31.0 °C), the window was heated in a controlled manner, as described in the previous section (Figure 7B). The content release from the TSLs was found to be tightly controlled, as fluorescent intensity was visible after heating for 90 s (39.5 °C). Drug release continued to increase during the heating period, with maximum extravasation and release visible at 510 s (42.1 °C). The discontinuation of hyperthermia application promptly ceased triggered release from TSLs, resulting in a decrease in fluorescent intensity within the tumor tissue.

### 3.2. Cylindrical RF Device (CRF)

The development of our novel radiofrequency-based device was based on a number of criteria. First, it required an adjustable focus location of the heated area to allow for the targeting and thermal sparing of specific regions. Second, with the intention to implement simultaneous magnetic resonance thermometry, the device had to be assembled from MR-compatible materials. Third, to allow for real-time intravital observations, the design had to reflect the desire to incorporate this feature.

The aim of the current model was to define a SAR suitable for heating a single tumor located in the liver. The focus of the heating device was a single SAR pattern of a defined size. The SAR payload parameters were influenced by the center frequency (f_c_) and the angle (Ψ). The center frequency determined the number of SAR foci. The angle was determined by symmetrically rotating both coaxial cables in relation to the frontal (X,Y) plane of the device, which allowed for the steering of the SAR (Figure 9A). Through SEMCAD X modelling, the optimal central frequency was set at 433.92 MHz and the optimal angle was set at 60°.

The presence of an internal water bolus strengthens EM coupling and minimizes spatial variation in tissue temperatures [27]. The design of our device required a second, smaller water bolus within the view port, as intravital image acquisition occurred through a water dipping objective. This feature was not only relevant to create high-resolution images of the microcirculatory systems using a confocal-multiphoton (SP5, Leica Microsystems, Wetzlar, Germany), but also affected the thermal boundaries. We observed that the addition of salt water (2.5 g/L) had a positive effect on the heat focus in heating measurements using a phantom. Moreover, using a water bolus without salt resulted in a large thermal zone throughout the whole phantom, as apparent in infrared images (Appendix A). The addition of salt to the water bolus within the view port not only led to a more defined thermal focus but also simultaneously moved the focus upwards in the direction of the liver and other superficial abdominal organs (Appendix A). Therefore, the adjustment of the water bolus with salt contributed to minimizing the risk of thermal toxicity in sensitive tissues such as the central nervous system (CNS).

The normalized SAR and temperature distribution patterns were predicted using SEMCAD X modelling software both for the phantom and in vivo models (Figure 10). The thermal images showed the transverse (X,Z) and sagittal (Y,Z) planes of each measurement. The black oval represented the position of the tumor in the model. The oblong shape was intentionally selected to simulate non-optimal conditions, focusing on a rather large and uneven volume.

In Figure 10A, the predicted SAR pattern for the phantom is shown. The applied RF settings generated an extended SAR region in the upper sagittal area of the phantom. The phantom temperature distribution pattern showed similar dimensions to the SAR region, with an observed temperature increase within the selected tumor region up to ΔT = 7 °C (Figure 10B). 

The in vivo simulations showed the generation of multiple SAR foci throughout the entire body, notably in the small and large intestine (Figure 10C). The temperature distribution pattern confirmed that the target temperature (T = 43.5 °C) was reached in the tumor region (P_abs_ = 27.9 W; t_input_ = 57 s) (Figure 10D). However, temperature elevations up to 45.0 °C were also reached in undesirable regions such as the intestines. A reduction of the P_input_, however, did minimize risk on thermal toxicity and other undesired effects, though it also increased the duration needed to reach target temperature.

Figure 11 shows in vivo profiles of the targeted tissue, core body, and spinal cord temperatures in a non-tumor bearing animal. The application of low power RF (P_input_ = 20 W; t = 30 s) resulted in stable temperature profiles for all three measured locations. Increasing RF power (P_input_ = 40 W; t = 30 s) returned similar profiles at a higher temperature.

### 3.3. Directional Microwave Antenna (DMA)

The directional microwave antenna model studied here was based on the extracorporeal visualization of a solid tumor located in the liver. In order to determine the feasibility of this device, preliminary studies were performed with the use of tissue-equivalent gel phantoms prior to moving forward to in vivo models. The visualization of the location and geometry of the generated temperature profile through infrared images showed that regions in direct contact with the applicator surface exhibited a lower temperature elevation than the more distant tumor regions (Figure 12B,C). This was in accordance with the simulated temperature profiles (Figure 12A).

In vivo microwave heating experiments demonstrated that local hyperthermia was established within a short timeframe. As shown in Figure 13, within 20 s after the onset of heating, the temperature elevation could be registered by the infrared camera. Figure 14 displays the tissue temperatures as measured by the thermocouples. Initial heating at a constant input power (P_input_ = 20 W; t = 10 s) resulted in a steep increase in tissue temperature (1.55 °C/s). The built-in safety metrics of the closed-loop feedback system ensured the modulation of the input power in order to avoid the overshoot of the set target temperature (42.0 °C). Figure 14 demonstrates that the feedback control loop induced a power drop to an average of 12.36 W (7.98–17.25 W) upon reaching 40.0 °C. The targeted area reached T_target_ within one minute and remained stable throughout the duration of the experiment (42.0 ± 0.2 °C).

To establish in vivo functionality, we performed the simultaneous intravital visualization of non-malignant hepatic tissue using the DMA set-up. The application of mild hyperthermia using the directional microwave hyperthermia system triggered local drug release (carboxyfluorescein) from TSLs, followed by the intratumoral distribution of the dye (Figure 15). The carboxyfluorescein-loaded TSLs circulated through the hepatic microvasculature (t = 0 min; pre-HT), and upon the initiation of hyperthermia application, the intravascular intensity became more diffuse while the extravascular intensity increased.

## 4. Discussion

A wide variety of devices for the applications of oncological hyperthermia, ranging from electromagnetic to ultrasound heating, is available, [31]. The main challenge of the currently available HT techniques is temperature homogeneity. While many studies have been performed in vitro and in silico, demonstrating the thermal effect in vivo is of a higher level of complexity. The hyperthermia devices available for preclinical studies are able to provide answers to the current challenges while accommodating (i) micro-environmental kinetics, (ii) precise control over heating quality, and (iii) improved understanding of how to evaluate hyperthermia-mediated mechanisms.

In this article, we describe three distinct preclinical models based on different heating methods that allow for the simultaneous high-resolution imaging of intratumoral events (Appendix A). The strength of the dorsal skinfold chamber lies in the accessible optical visualization. Tumor development can be visualized at all times in contrast to the other described models, which both require surgical intervention for unassisted vision of the deep-seated tumors. However, the CRF and DMA devices can be implemented to test more clinically relevant tumors. All of the described devices are able to heat a distinct volume to mild hyperthermic temperatures.

### 4.1. Micro-Environmental Kinetics

Thermal treatment triggers a dynamic tissue response by affecting five micro-environmental factors: (i) perfusion, (ii) permeability, (iii) pO_2_, (iv) pH, and (v) (intratumoral) pressure. These factors influence the tumor’s response to heating, and are, in turn, also altered by a local increase in temperature [3]. The extent of the response is dependent on the nature, size, and location of the tumor. On the one hand, the dynamic response could lead to differences in energy absorption in well-vascularized tumors. On the other hand, vasodilation could result in an increase in both fluid flow and vascular permeability. This may subsequently lead to synergistic cytotoxicity due to both enhanced drug delivery and sensitization caused by increased oxygenation [1,32]. These features have been omitted in simulations or phantom studies. Therefore, there is a need for better insight into the kinetics of heat application during preclinical studies [9].

Intravital microscopy, certainly when using the dorsal window chamber, comes with possible complications in addition to offering high resolution imaging. When combined with hyperthermia, extra attention should be paid to potential risks. One should keep the artificial setting in mind when interpreting observations. The DSC model provides a confined space in which a tumor grows and pressure build-up may occur. In addition, the positioning outside of the body may affect tumor growth, response, and intratumoral kinetics. Therefore, both temperature and humidity in the housing environment should be carefully maintained during tumor development. The application of hyperthermia may cause the expansion of tissue, change in fluid flow, and change in vessel performance. The additional build-up of pressure may occur. In our DSC model, we provided space for the tumor to grow and expand, and we closely monitored blood flow. However, the direct effect of HT may be disturbed by the effects of change in pressure, so when interpreting results, it is advised to confirm observations in a different setup.

The temperature range for mild hyperthermia in our study was set at 41.0–43.0 °C, as clinical HT procedures employ the same range and temperatures above 43.0 °C can cause irreversible vascular damage [10,33]. An important factor to take into consideration for treatment planning is the baseline temperature of the tumor. Temperature measurements within the dorsal skinfold chamber model showed a baseline tumor temperature of approximately 30.0 °C. At this temperature, none of the tumors exhibited liposome extravasation. However, increasing the temperature to 39.5 °C, which is at the bottom end of mild hyperthermia, resulted in nanoparticle extravasation in murine B16BL6 melanoma (Figure 8B). This occurrence supports the hypothesis that, in order to induce extravasation, the temperature elevation (ΔT) should be relative to the tumor’s baseline temperature. Therefore, a lower intratumoral temperature may be sufficient to induce tumor cell sensitization, enhanced extravasation, and site-specific triggered drug release within the dorsal skinfold window [34]. Further research should be conducted to prove this hypothesis.

As shown in the DSC model, the hyperthermia-induced local tissue temperature increased gradually. However, the discontinuation of heat application immediately induced a cool down effect (Figure 7B). This could have been induced by thermal convection into the dorsal skinfold window environment. It is likely that the loss of heat in the other two described hyperthermia devices more gradually occurred due to the presence of a higher temperature surrounding the target area. However, a decrease in tissue temperature did not limit the hyperthermia-mediated effects. The application of HT induced a sustained vascular permeability. While the highest degree of permeability may be expected during heat application, enhanced extravasation has still been observed up to 6–8 h post-HT [10,11,34]. It should be noted that the application of mild hyperthermia does not affect the functionality of the vasculature, and leakiness does drop back to the level observed before the occurrence of hyperthermia-mediated extravasation due to reversible gap formation (Figure 8B) [10].

### 4.2. Control

Hyperthermia treatment should ideally result in a homogeneous heat distribution, which is contributed to by factors such as tumor depth, volume, and location, which also affect therapeutic outcome [35]. Therefore, preclinical hyperthermia devices should be able to control these parameters via the controllable positioning of the heating element.

In our dorsal skinfold chamber model, the PEEK frames were designed to ensure the proper placement of the heating element using interlocking grooves (Figure 2C), and prior to microscopic imaging, the window is secured to a chamber-to-stage holder using bolts that fit the two large recesses in the casing of the heating element (Figure 2D). This not only ensures perfect alignment but also enhances window stabilization and repeatability during longitudinal studies.

In the cylindrical RF device, control occurs through the mechanical steering of the heat focus (Figure 9A). In the current model, the set angle (Ψ = 60°) resulted in a SAR pattern with clear thermal boundaries. Phantom simulations indicated that energy deposition solely occurred at the site of interest, circumventing areas sensitive to thermal toxicity (Figure 10A). However, it should be noted that the dimensions of the simulated SAR and temperature patterns were larger than the indicated tumor diameter. Despite being able to avoid heating of the CNS, unwanted heat deposition may have occurred at other sites. Further research should be performed to optimize the device for the minimal heating of non-malignant tissues.

In the directional microwave hyperthermia device, the placement of the antenna is effortlessly controllable due to the design of the set up (Figure 4C). As the animal is always placed in the same position, the movement of the target tissue along the x,y,z axis is limited. The flexible arm has an adjustable angle of 360° and can therefore also be placed at an incline in order to ensure the complete coverage of the target area. Hyperthermic control also occurs through the presence of circulating water throughout the device. Simulated temperature profiles in an agar phantom indicated that the water bolus enables the thermal sparing of the superficial layer (Figure 12A) [29]. This aspect was also visible in the in vivo studies, as the infrared images indicated a lower temperature for regions in direct contact with the applicator than the more distant regions (Figure 12B). This is a beneficial feature of this hyperthermia device, as some studies have reported acute or chronic damage to the skin at temperatures above 41.0 °C [33].

Thermometry is also an essential factor in the control of hyperthermia devices. During thermal treatment, the internal and external temperatures should be monitored at all times. While the rectal and skin surface temperatures are important for the welfare of the animal, intratumoral temperature should be monitored to confirm proper targeting and to ensure that the set temperature is reached but not exceeded [36,37]. A conventional method for thermometry is with the use of multiple thermocouples inserted at various sites to roughly determine the temperature distribution pattern. For the three hyperthermia devices discussed in this paper, we used thermocouples for preliminary studies. However, the invasiveness of this method should be considered, as probe insertion leads to parenchymal disruption and may create artefacts in tumor circulation [38]. More advanced systems may focus on the inclusion of non-invasive techniques such as infrared (IR) or magnetic resonance thermometry (MRT). However, it should be considered that these techniques may not provide the same accuracy as thermocouples due to tissue dependence and susceptibility to motion artifacts [31,39].

Thermometry does not solely provide information on heat distribution in tissues; incorporating thermal feedback to the hyperthermia device is also beneficial for controlling the output power and, thus, the quality of heating. The design of the cylindrical CRF device and the directional microwave antenna system includes feedback of the applicator performance to the power generator through thermocouples. At the onset of HT application, the maximum set power is applied. The software developed for this device continuously provides feedback to the amplifier depending on the values of the PID controller. Therefore, upon approximating the target temperature in the measured region, the output power diminishes (Figure 14). Until the set temperature has been reached, the amplifier provides a low power to gradually increase the intratumor temperature to mild HT.

The precise control of the heating devices is essential to keep the tissue temperatures in the mild hyperthermic range (40.0–43.0 °C). In the DSC model, the in vivo temperature profiles indicated that the temperature of the tumor and surrounding tissue were similar. This may not be an obstacle, as long as the temperatures remain below 43.0 °C in order to prevent detrimental effects such as blood flow stasis, vascular collapse, and hemorrhage [10,40,41]. In the study using the CRF device, both core and spinal cord temperatures displayed a slight increase in comparison to the lower applied input power, as all measurements were performed in a single mouse without a lengthy cool down time. The P_input_ of 20 and 40 W corresponded to measured forwarded power values of 5 and 10 W, respectively. Therefore, the device should be tuned in order to reach stable hyperthermia application. The inclusion of advanced thermographic measurements is required to ensure proper thermal treatment.

### 4.3. Complex Evaluation

The evaluation of biological and physiological events occurring before, during, and after hyperthermia treatment is complex. It is desirable to visualize the kinetics in high spatial and temporal resolution. Imaging modalities such as MRI are clinically relevant and non-invasive. However, the spatio-temporal resolution of MRI is not optimal for the real-time imaging of (sub) cellular dynamics [42]. Optical imaging modalities, such as confocal or multiphoton microscopy, can be simultaneously operated with the hyperthermia devices to obtain high-definition images. Therefore, IVM implementation yields both structural and functional information, elucidating, e.g., nanoparticle accumulation, tumor uptake, and longitudinal therapeutic response under the influence of mild HT [17,43].

The IVM visualization of HT-mediated events can be achieved with the concomitant administration of liposomes. The dorsal skinfold chamber model can be performed in animals expressing fluorescent proteins, as we have shown here for the vascular endothelial cell layer. This enables the clear distinction between tumor vasculature (green) and circulating liposomes (red) (Figure 8A). Upon the local application of hyperthermia, the dissemination and intensity may change due to vasculature extravasation or triggered liposomal content release (Figure 8 and Figure 14). Further improvements of evaluation in the DSC model can be made with the use of a photo-etched cover slip on the tumor side of the window. This structure contains a permanent grid pattern to allow for the imaging of the same position throughout longitudinal experiments [44].

High technical complexity applies to all devices, as these combine advanced technologies and demanding mouse models. If this is undesired, especially for the physiological and biological evaluation of superficial tumors, the implementation of other hyperthermia devices is recommended [16]. The main drawback of the dorsal skinfold chamber is the previously mentioned difference in baseline tumor temperature. This may be a minor influence in non-thermal studies as long as the animals are housed at a proper ambient temperature, which is (in our experience) 32.0 °C [25]. For the cylindrical RF device, the presence of a water bolus improves both coupling and thermoregulation. However, this also makes the device vulnerable because the lining of the water bolus is prone to rupture. For the directional microwave device, the main inconvenience is the exact placement of the antenna. In order to overcome this problem, an infrared camera may be incorporated in the set-up to provide insight into the temperature distribution and dynamics based on the superficial tissue temperatures (Figure 12C). To make the organ accessible for microscopic visualization, the liver lobe must be exteriorized and carefully positioned onto the glass microscope slide. This requires not only invasive surgery but also organ manipulation from the anatomical position [45]. This procedure is further optimized in order to allow for the execution of longitudinal studies.

## 5. Conclusions

The application of mild hyperthermia displays a plethora of effects in and around the heated area. These can affect tumor progression, response to (co)therapy, and intratumoral pathophysiology. Moreover, hyperthermia-controlled therapies, for instance with thermosensitive liposomes, have complex intratumor kinetics in both time and space. To better understand the effect of mild hyperthermia, alone or when used in combination, detailed insight is needed. The use of devices that enable the application of hyperthermia to experimental tumors and simultaneous high resolution optical imaging provides powerful tools to dissect the abovementioned processes. The set-ups discussed here show imaging up to the subcellular level before, during, and after application of controlled hyperthermia and can deliver answers to further delineate the involved processes.

## Figures and Tables

**Figure 1 cancers-13-05146-f001:**
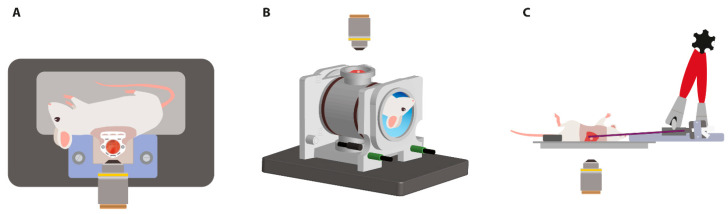
Illustrations of the devices for the application of local hyperthermia treatment in combination with intravital imaging. HT application occurs through (**A**) an external heating element adjacent to the dorsal skinfold chamber (Figure 2), (**B**) the generation of an oscillating electromagnetic field (Figure 3), and (**C**) the placement of the directional microwave hyperthermia system (Figure 4).

**Figure 2 cancers-13-05146-f002:**
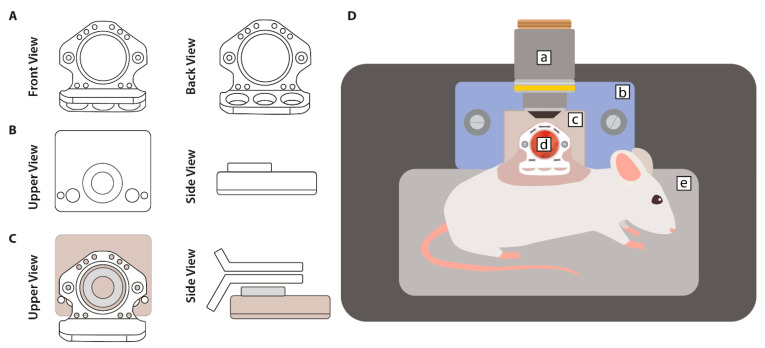
Illustrations of the dorsal skinfold chamber (DSC) model. (**A**) The window model consisted of two non-symmetrical frames that are made of polyether ether ketone (PEEK); (**B**) local hyperthermia was provided by an external circular resistive electric heating coil; (**C**) the window was secured to the heating element using screws in combination with the chamber-to-stage platform on the microscope; (**D**) the DSC model set-up consisted of (a) microscope objective, (b) chamber-to-stage holder, (c) external heating element, (d) dorsal skinfold chamber, and (e) heating platform for the maintenance of the core body temperature of the animal.

**Figure 3 cancers-13-05146-f003:**
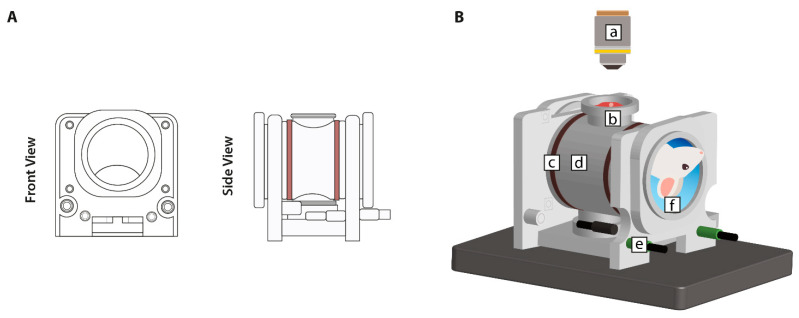
Illustrations of the cylindrical RF device for deep hyperthermia. (**A**) The device consisted of a 3D printed cylindrical base with a hollow bore; semi-circle copper antennas (brown) generated an electromagnetic field, which induced a local temperature increase; (**B**) the CRF set-up consisted of (a) microscope objective, (b) view port, (c) antennas, (d) cylindrical 3D printed frame, (e) coaxial cables, and (f) water bolus.

**Figure 4 cancers-13-05146-f004:**
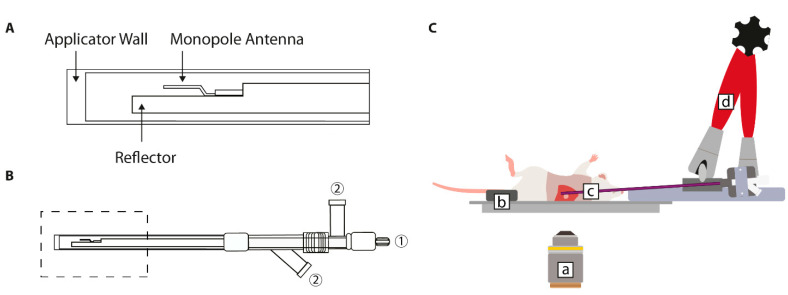
Illustrations of the hyperthermia model using a directional microwave hyperthermia antenna. (**A**) The radiating part of the applicator consisted of a coaxial monopole cable, which was surrounded by a copper outer conductor. At the end of the monopole, the outer conductor was shaped as a reflector by adopting a hemicylindrical form. The outer wall consisted of polyimide, which encapsulated the water circulating through the applicator. (**B**) The marked section indicates the applicator part as shown in A. Three connectors were located on the other side of the device: one for the RF power ① and two for the water circulation ②. (**C**) The directional microwave hyperthermia antenna set-up consisted of (a) microscope objective, (b) heating platform for the maintenance of the animal’s core body temperature, (c) coaxial monopole antenna, and (d) flexible arm with 360° rotation.

**Figure 5 cancers-13-05146-f005:**
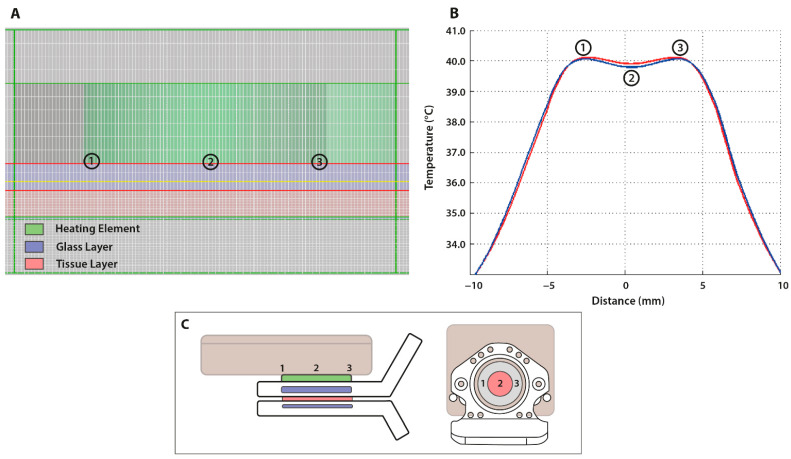
SEMCAD X simulations of the temperature distribution within the DSC heating element. (**A**) Image from the SEMCAD X software showing the simulated DSC model, which consisted of a heating element (green) and the skinfold chamber window being a combination of tissue (red) sandwiched between two glass plates (blue); (**B**) the temperature profile simulated at the border between the heating element (green) and the lower glass plate (blue); (**C**) illustrations of the dorsal skinfold chamber model in relation to the SEMCAD X simulations. The edges of the heating element (point 1 and 3) are shown to have been indirect contact with the glass layer, while the hollow bore allowed for transillumination (point 2).

**Figure 6 cancers-13-05146-f006:**
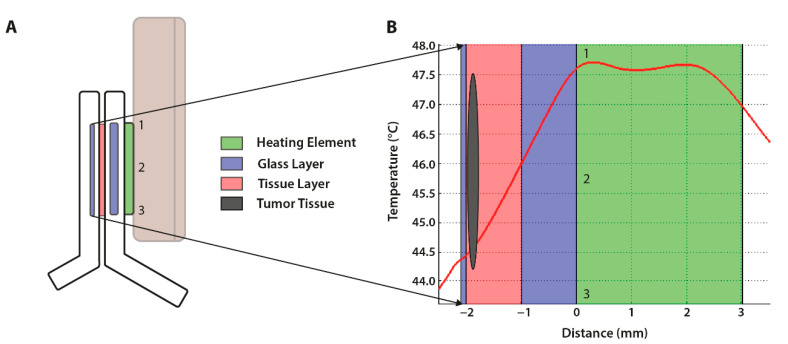
Schematic representation of the dorsal skinfold chamber (DSC) model. (**A**) The dorsal skinfold chamber was secured to the heating element. As shown, point 1 and 3 of the heating element (green) were in direct contact with the glass bottom layer (blue). Point 2 yielded no direct thermal conduction, as the hollow bore of the element was located at that site. (**B**) The SEMCAD X simulated temperature profile of the DSC model. The temperature within the heating element (green) approximated the set temperature. Upon conduction into the bottom glass layer (blue), the temperature decreased and continued to decline as heat transferred into the tissue layer (red), upper glass layer (blue), and air (white). The SEMCAD X simulation indicated that the device temperature had to be set at ±47.5 °C in order to reach a tissue temperature of ±44.5 °C (black oval).

**Figure 7 cancers-13-05146-f007:**
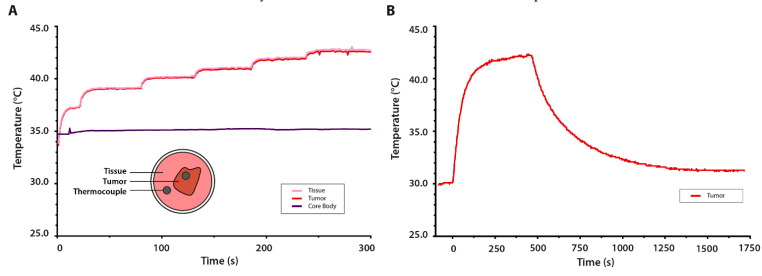
In vivo temperature measurements in the dorsal skinfold chamber. (**A**) Temperature distribution in the window at multiple locations during interval heating; the temperature was manually set 1 degree Celsius higher per interval after first reaching 37.0 °C. (**B**) The intratumoral temperature during mild hyperthermia application; the temperature increased from steady state temperature in the window to the set target temperature of 42.0 °C, after which the power was shut down to track the cool down.

**Figure 8 cancers-13-05146-f008:**
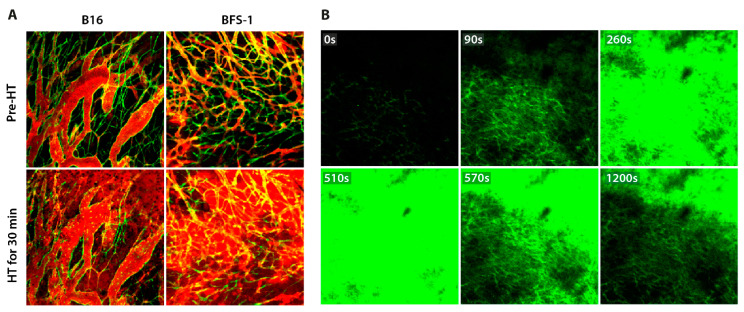
Hyperthermia (HT)-mediated intratumoral liposomal drug delivery. (**A**) HT (41.0 °C; 30 min) triggered rhodamine-labelled long circulating liposome (red) extravasation through murine B16BL6 melanoma or murine BFS-1 fibrosarcoma tumor vasculature (green). At physiological temperature, extravasation hardly occurred. HT application increased vascular permeability and enabled the extravasation of liposomes. (**B**) HT (42.2 °C; 5 min) triggered drug release (green) from thermosensitive liposomes (TSLs) in B16BL6 murine melanoma. TSLs released content (carboxyfluorescein; CF) when exposed to HT (39.5 °C or higher), resulting in rapid distribution of CF throughout the tumor. However, TSLs appeared stable at a physiological temperature and no CF was released when the intratumor temperature returned to baseline.

**Figure 9 cancers-13-05146-f009:**
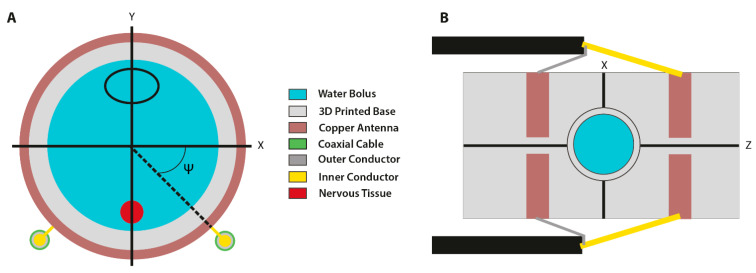
Schematic representation of the cylindrical RF device in (**A**) the X,Y plane and (**B**) the X,Z plane. (**A**) The generation of an oscillating electromagnetic field occurred through the inner/outer connections of the coaxial cable to the semi-circle antennas. The black oval represents the position of the tumor in the model. The red filled circle indicates the position of the central nervous system. Angle Ψ is the optimal angle to steer the heat focus towards the tumor. (**B**) Upper view of the CRF device. The view port was located in the middle of the device, between the four semi-circle copper antennas.

**Figure 10 cancers-13-05146-f010:**
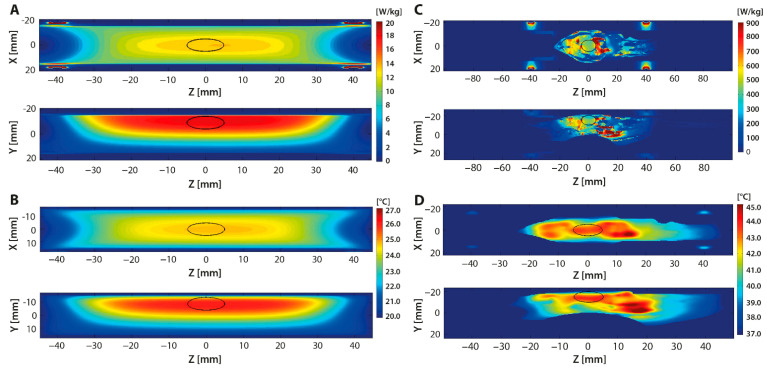
The specific absorption rate (SAR) and temperature distribution patterns of a homogeneous phantom (**A**,**B**) and mouse model (**C**,**D**) were simulated at fc = 433.92 MHz and Ψ = 60°. For the temperature distribution patterns, the following settings were applied: (**B**) P_input_ = 20 W and t = 60 s; (**D**) P_input_ = 34 W and t = 57 s. The thermoradiographs show the transverse (X,Z) and sagittal (Y,Z) planes of each measurement. The black oval represents the position of the tumor in the model.

**Figure 11 cancers-13-05146-f011:**
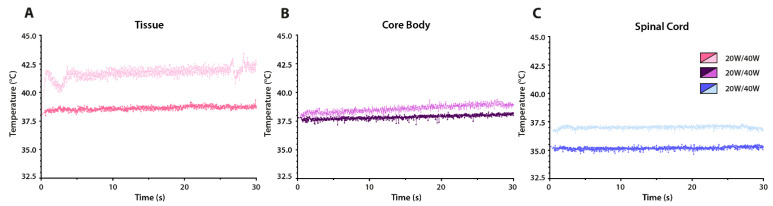
In vivo temperature profiles in the cylindrical heating device based on two input powers (20 and 40 W) for 30 s. (**A**) The heating profiles as measured with thermocouples within the target tissue; (**B**) temperature profiles within the core body; and (**C**) temperature profiles near the spinal cord.

**Figure 12 cancers-13-05146-f012:**
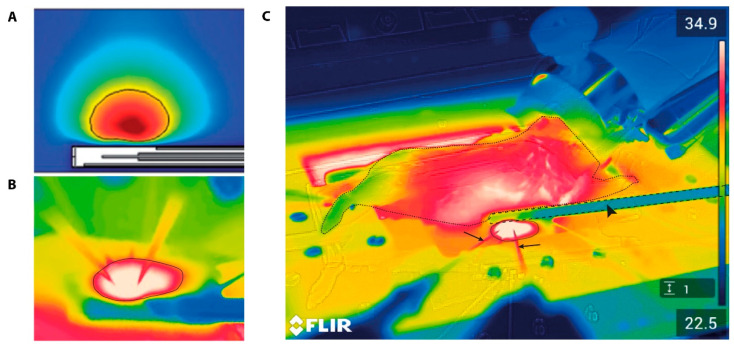
Simulated (**A**) and in vivo (**B**,**C**) temperature profiles of the DMA. (**A**) Simulated temperature profiles of the device indicate that the hot spot was directly in line with the monopole antenna. In addition, regions in direct contact with the applicator surface had a diminished temperature elevation in comparison to the more distant tumor regions; (**B**) in vivo temperature profiles are similar to the simulations from (**A**), indicating the lowest temperature in the water-cooled antenna (blue) and the highest temperature in the target area (white); (**C**) infrared image of the set-up overview: the animal is marked by the thin dashed line, the antenna is marked by a dashed line and indicated with a closed arrow head, the liver is marked with a solid line, and the arrows indicate the needle-tip thermocouples. Figure 12A is reproduced with permission from Curto et al. (2018) [29].

**Figure 13 cancers-13-05146-f013:**
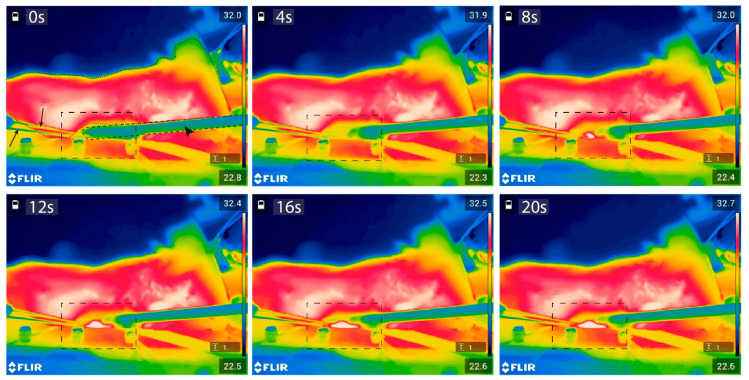
Infrared images of the in vivo region of interest during initial hyperthermia application. In panel 1, the animal is marked by the thin dashed line, the antenna is marked by a dashed line and indicated with a closed arrow head, and the arrows indicate the needle-tip thermometers. The region of interest is marked in each panel by the dashed box.

**Figure 14 cancers-13-05146-f014:**
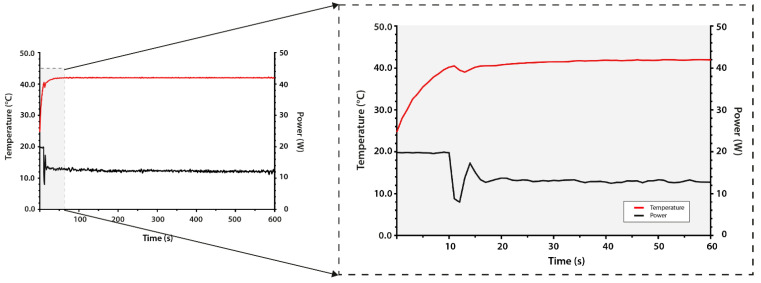
Temperature and power as a function of exposure time using the directional microwave hyperthermia device. The insert demonstrates the build-in feedback loop; upon reaching a ΔT < 2.0 °C from the target temperature (Ttarget, 42.0 °C), the power decreased. This subsequently led to a gradual increase to the Ttarget.

**Figure 15 cancers-13-05146-f015:**
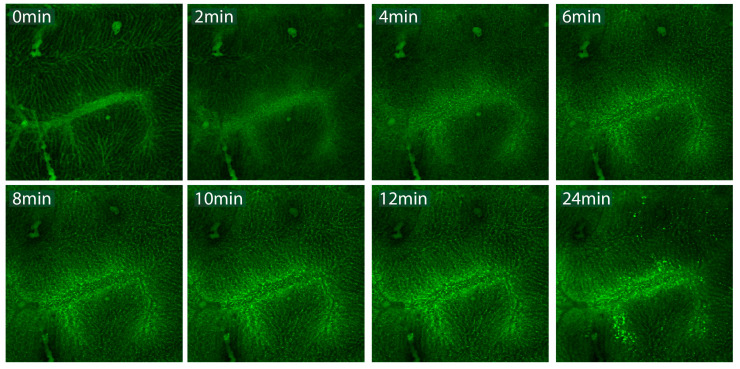
Hyperthermia-mediated intrahepatic liposomal drug delivery. At baseline temperature, carboxyfluorescein-loaded thermosensitive liposomes circulated through the hepatic microvasculature (t = 0 min). Subsequent HT application (41.0 °C; 24 min) gradually triggered content release from TSLs, resulting in the distribution of free CF, as shown by the increased extravascular intensity.

## Data Availability

The data presented in this study are available in the article and supplementary material.

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
