# Peer review of "Preclinical Studies in Small Animals for Advanced Drug Delivery Using Hyperthermia and Intravital Microscopy"

_cancers, 2021, doi:10.3390/cancers13205146_

Round 1

Reviewer 1 Report

Using the built-in intravital microscope and thermosensitive, fluorescent dye-loaded liposomes, the authors evaluated and compared the 3 known heating devices for the application of hyperthermia in mice. They made some interesting observations such that release of the dye during heat treatment was promptly ceased by strict discontinuation of hyperthermia application. However, there are a few issues.

  1. Abstract: This is not informative and needs to rewrite. Too long Introductory remarks in "Simple Summary" (Line 1-7) and in "Abstract" (Line 1-10). Instead, the authors should include important results obtained, and brief discussion
  2. Results and Discussion P17: End of the first paragraph. They stated that “To induce extravasation, the temperature elevation (ΔT) should be relative to the tumor’s baseline temperature” and suggested that there is no absolute thermal dose. For justification of this statement, they should test changing the baseline temperature (for examples to 28 C and 32 C) and measure extravasation temperature. Confirmation is necessary if the thermosensitive liposomes themselves are not affected  by the temperature shift.
  3. Discussion P17: Paragraph 4, 5 and 6. These statements are just simple repetition of “Methods” and tedious.
  4. References 13 and 27: No information is shown about Advanced Drug Delivery Reviews.

Reviewer 2 Report

Authors introduced three different devices used for the application of hyperthermia in small animal models (mice). These heating devices allow for intravital microscopy for real-time visualization of intratumoral events, such as drug delivery. These devices include the external electric heating of the dorsal skinfold model, targeted radiofrequency application (oscillating electromagnetic field), and microwave antenna heating.

Limitations of the study were very well described; however, the manuscript may benefit from several minor additions and clarifications.

Please see the comments below.

Concern 1: Please provide more detail on the method of “enzymatic detachment” of the cell lines, eg trypsin.

Concern 2: Pre-op and post-op analgesia is described for the tumor implantation studies. Were the same analgesic agents used for the surgeries for exteriorizing the liver? Were these studies terminal or survival? If terminal, please describe the method of euthanasia.

Concern 3: Is the department of Experimental Medical Instrumentation located at the same institution?

Concern 4: For the dorsal skin fold model, please explain why a second glass coverslip was used on the back for the heating and not heating directly on the tissue.

Concern 5: Please describe the anesthesia used during imaging/heating. Isoflurane use is mentioned for tumor implantation. Was the same used for imaging?

Concern 6: Please clarify the imaging modality used for the CRF device for the tumor imaging/drug delivery. Is the liver tissue exposed in this model? The article states that high-resolution imaging of intratumoral events can be performed; however, these images are not provided for the CRF device.

Concern 7: Data/images provided in Figure 8 are similar to previously published work. Please cite previous references for these data (eg Manzoor and Lindner et al Cancer Research 2012).

Concern 8: Page 12, second paragraph, Please define SAR the first time it is abbreviated.

Concern 9: Why were tumor-bearing mice not used in the CRF device study?

Concern 10: Can images of the goal tumor temperature of 42C be provided in Figure 13?

Concern 11: Please clarify whether Figure 15 shows tumor tissue or normal hepatic tissue. If the image includes tumor tissue, can the tumor tissue be outlined in the images?

Concern 12: Please provide a reference for the information provided in the last sentence of paragraph 2 on page 17.

Reviewer 3 Report

General comments: The authors present data on the impact of HT delivered by different modalities on tumors grown in a non-compliant space. It is well known that during mild HT tissue edema develops that even more complicates this situation. Thus, this tumor model is far away from reality in the clinical setting. A series of critical items are not discussed in this article.

Specific comments: 

In the Introduction section an inadequate selection of references is presented: DSC have been used in HT research before. These should be referenced. Tumor physiology upon HT is more complex than described in [3]/first para of the Introduction.  Complexity of the situation has been published before. Therefore, "imperatives"should be avoided in the 2nd para (-can, could, may, might- should be used instead).

In the methods and results section there is a plethora of technical details that  should be published in a separate, more technically oriented journal.  This part of the article is completely "overloaded".

In the discussion part, the authors describe phenomena which certainly are

lacking in the tissue of malignant tumors (especially in tumors growing in the highly artificial preparation described here). These include the following statements: "homeostatic regulation", "physiological feed-back", "physiological regulation" etc.

Round 2

Reviewer 1 Report

References 13 and 27: volume and page numbers are missing.

13.  -----2020;163-164:125-144.

27.     -----2020;163-164:3-18.

Author Response

We would like to thank you for pointing this out. We have updates the references in the latest version.

This manuscript is a resubmission of an earlier submission. The following is a list of the peer review reports and author responses from that submission.

Round 1

Reviewer 1 Report

The manuscript by Priester et al. provides an extensive review of external hyperthermia devices for preclinical studies in small animals. Section 3 on hyperthermia devices without optical monitoring provides an up-to-date and extensive overview of the different techniques and cites all relevant papers by different groups. However, section 4 on hyperthermia devices with simultaneous biological monitoring reads like a collection of unpublished preliminary experiments performed in their group. Therefore, this section is not on topic, contains a lot of non-relevant data and does not provide an objective view on the field. I will concretize this below in my detailed comments.

Major comments:

Water bath heating (line 392-405): As the authors mention with water bath heating it is impossible to not heat the surrounding healthy tissue. However, I am missing a discussing on the physiologic effect in the tumor (mainly perfusion) when heating the surrounding tissue. And how does this compare to other heating methods (e.g. HIFU) that allow for tumor-only heating (at least in larger animals and humans)?

HIFU (line 536-617): This paragraph contains many mistakes and needs thorough revision. Here follows a list of issues that at least should be tackled:

  • Line 548: “therapeutic ultrasound is able to localize … the tumor …”. With ultrasound imaging it is possible to localize a tumor, with therapeutic ultrasound you can treat a tumor. Two times ultrasound, but for sure not used in the same way.
  • Line 553-563: the authors mix up imaging transducers and therapy transducer. Single and split beam transducer is related to therapy, while linear and phased array transducers are related to imaging
  • Line 570-571: “short duty cycles” don’t exist, duty cycles are low or high. In addition, the duty cycle is determined by the pulse length and the pulse repetition frequency. So please rephrase.
  • Line 573-576: the PRF is not equal to frequency. The PRF does not determine penetration. Please correct
  • Line 576-577: the applied thermal dose is not directly determined by the treatment duration. After the thermal treatment has stopped the tissue temperature can remain elevated and consequently the thermal dose continues to increase.
  • Line 589-591: I really don’t understand how the water pressure determines the depth of focus. Please explain.
  • Line 598-600: the authors imply that all “HIFU therapeutic transducers transmit focused ultrasound energy in concentric or ellipsoidal circles”, but that’s not correct. The natural shape of a focal point is something between a sphere and a ‘cigar-shape’ (depending on the geometry of the transducer). Subsequently, this natural focal spot can be moved via electronic beam steering in e.g. concentric circles.

Redundant/not on topic paragraphs: Section 4 contains many paragraphs that describe in detail the construction of e.g. a dorsal skinfold window. This might be a nice topic for technical note, though does not fit well in this review paper. I suggest to shorten or completely remove the following paragraphs:

  • 1 DSW: this should be written in the same manner as the previous devices (i.e. water bath, cold light etc) are discussed and leave out all the technical details, simulations etc.
  • Remove completely lines 868-882: this paragraph on temperature triggered drug release does not fit in the DSW paragraph.
  • 3 CDA, line 1136-1174: these paragraphs on tumor inoculation and growth do not fit with the rest of the paragraph

Self-reference: Although, in the first part of this manuscript many relevant papers of other groups are cited, the amount of self-reference is disproportional.

Minor comments:

  • Line 280-285: why subtracting only the ROI? In general, for PRFS thermometry a reference image (before) heating is subtracted from the images acquired during heating. There is no need for an ROI. When comparing MR thermometry with temperature probes It is also good to mention that the PRFS technique only measures temperature changes and not the absolute temperature.
  • Line 405: the authors suggest that “a large water bolus” always hampers MR imaging, but this is not true. The presence of a water bolus can help the shimming of the MR system, though sometimes the water has to be doped to reduce the signal.
  • Line 688: In my opinion HIFU experiments also “…require advanced skills and thorough training before reproducible results can be obtained”, so maybe the authors could rephrase this a bit less subjective.
  • Line 1330: what is meant with “..alignment of the three different heating methods…”?
  • Line 1334- 1346: this paragraph should not be part of the discussion section
  • Line 1351: unclear to which technique is referred with “…optimize the technique.”
  • Line 1352: unclear to which paper is referred with “… this paper …”

Reviewer 2 Report

This must have been a lot of work.

I am sorry to say that for me, as a reader, it remains bulky, loos information, without any clear objective, aim, or conclusion. I think the authors, specifically need to adress one subject. Choose a focus, for instance, small animal devices to be used in drug delivery experments. Due to the bulky information, specific data loose it's context and meaning throughout the paper. In and exclusion criteria need to be defined clearly. The discussion is not as it should be, yet another study is described. the conclusion is a mere repetition of the introduction, with a loose last sentence.

Reviewer 3 Report

The first three Chapters are extremely superficially covering the techniques.
This is in extreme contrast to Chapter 4 describing the authors approach in high detail. While Chapter 4 described the experimental setup in extreme detail including simulation. Hence, overall this is not a well-balanced review. Essentially it reads as two manuscripts. Chapter 1 to 3 and Chapter 4.

NIR heating easily allows for optical monitoring as in the dorsal skinfold window irradiation and monitoring can be from opposite sides.

Why not also present heating modelling for NIR or HIFU heating, the former based on diffusion therapy and bioheat transfer equation, whereas HIFU is subject to the thermal relaxation time of the focal volume. 

There are also various repetition (tumour inductions or excessive unnecessary information. For example line 143 ff about the murine genetic expression. What does this provide to the reader related to metastatic potential?

Be more critical for example the use of Thermocouples to measure temperature during NIR heating and also broadband white light heating is not advisable as the bimetal junction is absorbing light and will report higher temperatures but also locally heat, hence perturb the system.

The description of the Cold Light Source should emphasise that the emission profile follows Black body radiation.  No need to describe the function of a laser. Collimation and coherence are of no consequence for turbid media such as tissue.

LIne 425 when referring to the tissue optical window consider doi:10.1088/0031-9155/58/11/R37

LIne 443 please note that white cotton or fabrics do not prevent transmission of the light. This is misleading to the reader. Gauze can transmit light in the NIR range see  DOI:10.1089/clm.2000.18.235

Heating with 800 nm has haemoglobin as the dominant absorber but that is by no mean optimal. When introducing penetration depth, emphasise that this is the 1/e effective attenuation distance.

There is a persistent problem with the reference manager, please check your PDF prior to submission.

Recommendation, split into two manuscripts the one representing the dutch experience in pre-clinical hyperthermia and a critical discussion of other technique.

Reviewer 4 Report

Review of Manuscript Cancers-1194517

General Remarks

The manuscript presented is an excellently written review of small animal models for the study of hyperthermia with and without intravital microscopy. The presentation of the methods is excellent and provides a unique understandable overview and allows a detailed comparison of these methods in terms of advantages and disadvantages of each method, which has never been published in the scientific literature with the thoroughness and completeness provided.

Due to the outstanding quality of the manuscript, I can only congratulate the authors on their excellent work. I have no criticisms to make, nor have I been able to find any errors in the presentation of the methods. The individual sections are linguistically very well formulated and structured. The conclusions and recommendations from the review article are comprehensible and consistent.

I recommend the manuscript directly for publication without any change requests.

Specific Remarks

  • Line 101-102 Reference Error
  • Line 357 Reference Error
  • Line 437 Reference Error
  • Line 510 Reference Error
  • Line 539-540 Reference Error
  • Line 1023 Reference Error

Round 2

Reviewer 1 Report

In the current form I unable to review the authors' response to the reviewers. Action from editor is required.

Reviewer 2 Report

See previous comments. In addition, I find 43 pages much too long for a clear and specific review.

Reviewer 3 Report

The Authors have rewritten the manuscript to a large extent which includes high detail on particular animal models. Some of the hyperthermia delivering methods, in particular interstitial laser hyperthermia, previously insufficiently explained. While the quality of the review article has significantly improved the scope has been more focused, reflecting more the expertise of the authors. As this reviewer is not an expert in the presented hyperthermia techniques no notable shortcomings were noted.